# Two-Parametric, Mathematically Undisclosed Solitary Electron Holes and Their Evolution Equation

Hans Schamel 

Physikalisches Institut, Universität Bayreuth, D-95440 Bayreuth, Germany; hans.schamel@uni-bayreuth.de

**Abstract:** The examination of the mutual influence of the two main trapping scenarios, which are characterized by $B$ and $D$ and which in isolation yield the known $\mathrm{sech}^4$ ($D = 0$) and Gaussian ($B = 0$) electron holes, show generalized, two-parametric solitary wave solutions. This increases the variety of hole solutions considerably beyond the two cases previously discussed, but at the expense of their mathematical disclosure, since $\phi(x)$, the electrical wave potential, can no longer be expressed analytically by known functions. Therefore, they belong to a variety with a partially hidden mathematical background, a hitherto unexplored world of structure formation, the origin of which is the chaotic individual particle dynamics at resonance in the coherent wave particle interaction. A third trapping scenario $\Gamma$, being independent of $(B, D)$ and representing the perturbative trapping scenarios in lowest order, provides a broad, continuous band of associated phase velocities $v_0$. For structures propagating near $C_{SEA} = 1.307$, the *slow electron acoustic speed*, a Generalized Schamel equation is derived: $\varphi_\tau + [\mathcal{A} - B\frac{15}{8}\sqrt{\varphi} + D\ln\varphi]\varphi_x - \varphi_{xxx} = 0$, which governs their evolution. $\mathcal{A}$ is associated with the phase speed and $\tau := C_{SEA}t$ and $\varphi := \phi/\psi \geq 0$ are the renormalized time and electric potential, respectively, where $\psi$ is the amplitude of the structure.

**Keywords:** kinetic; nonlinear; trapping; separatrices; holes; chaos; intermittence

## 1. Introduction

In the last few decades, a plethora of noteworthy studies of very different character have been presented that deal with the development of the microscopic texture of a driven plasma in the high temperature, dilute density limit, i.e., with structure formation in collision-free plasmas in a time-dependent setting. Various aspects of the excitation as an initial value problem were addressed numerically and/or treated analytically. To mention three examples: the nonlinear frequency shift of an ion acoustic wave was studied in [1] as a function of a sudden or an adiabatic switching on of the disturbance, the existence and approach of non-Landau solutions as a result of a special preparation of the initial plasma state was addressed in [2], or the excitation of large amplitude structures, called KEEN waves (Kinetic Electrostatic Electron Nonlinear), being driven e.g., by the ponderomotive force of two crossing laser beams, was investigated in [3], with all of them representing valuable contributions to the solution of the riddle of kinetic structures. However, the above choice of three works is by no means mandatory, as other equally relevant works could be selected as well, such as [4–7].

However, the main problem associated with these studies is that there is theoretically no resolvable link between the initial state and the final asymptotic hole state. This is especially true if initially seed-like, non-topological fluctuations in the distribution are admitted [8–10]. In the transient transition stage filamentation, folding, trapping, detrapping or retrapping processes etc. occur which are too complex to be handeled analytically. Another reason for the lack of stringent non-stationary solutions of the full Vlasov-Poisson system is the non-integrability of the single particle-coherent wave interaction problem at resonance, which is reflected in the complexity of resonant characteristics of the Vlasov equation.

Mathematically speaking, hence, the task is hopelessly gigantic, because it boils down to finding suitable paths through the chaos paved with obstacles such as KAM theory, Arnold diffusion, separatrix crossings and pulsations or violent relaxations, to name just a few key words.

Fortunately, this is different for *stationary* solutions of the Vlasov-Poisson system, where for given trapping scenarios *exact* and *complete* solutions can be obtained due to the pseudo-potential method in the version of Schamel [11]. Most recent investigations [12–14] reveal that these coherent structures are (i) strictly nonlinear, no matter how small the amplitudes, (ii) that continuous rather than linear discrete phase velocities mark their speed, and (iii) that consequently there are an unlimited source of experimentally unidentifiable hole solutions.

How difficult this task is and what irritation can arise in its interpretation can already be seen in the single harmonic wave limit of hole structures. In the small amplitude limit a dispersion relation (DR) of the Thumb-Teardrop type can either be obtained linearly [15–18] or nonlinearly [11,19–24], the four branches of which refer to the Langmuir wave branch, the slow electron acoustic wave branch (SEAW), the ion acoustic wace branch (IAW), and the slow ion acoustic branch (SIAW). However, only two of them, the Langmuir and the ion sound branch, survive a validity check of the linearization procedure whereas all four are reliable solutions of the nonlinear system up to the infinitesimal amplitude limit. Hence, the Thumb-Teardrop DR is formally a linear one, but is, in reality, only justified as a nonlinear dispersion relation (NDR). The electron acoustic wave (EAW), being identical with the earlier termed SEAW [20–22], is hence by no means a linear wave in contrast to the current picture of this mode (see e.g., [17]). For more details, see Sect.IV and the controversial discussion in [18,25].

This SEAW—the same holds for the SIAW—plays a central role in Schamel's theory. It is not only the correct nonlinear extension of the linear zero-damped van Kampen and Landau mode, respectively, represented by the corresponding perturbative trapping scenario (namely B = D = 0, $\Gamma \neq 0$ in the first and B = D = 0 = $\Gamma$ in the second case, see later and [12]) but turns out linearly unconditionally marginally stable in a current-carrying plasma independent of the drift velocity $v_D$ and the temperature ratio $T_e/T_i$ [26] in stark contrast to Landau's theory [15].

Hence, its nonlinear character is kept in a strong microscopic sense up to the infinitesimal amplitude limit introducing generally speaking a gap or cut between the linear and nonlinear wave function space (see e.g., Figure 2 of [27]). This fact seems to be in agreement with Mouhot and Villani's assessment of linear and non-linear Landau theory (strong convergence macroscopically and weak convergence microscopically), as its proof strongly relies on the homogeneity and perturbative treatment of the problem [28–30].

The failure of Landau theory in case of coherency and non-smooth, seed-like initial conditions is evident in the series of papers published by Mandal, Schamel, and Sharma [8–10,12,13], where robust solitary electron holes (SEHs) were excited in the subcritical regime of a current-carrying plasma without any signature of damping. These structures are triggered by tiny, eddy-like, non-topolgical seed fluctuations. Their omnipresence without threshold values is an indication of a perturbative trapping scenario that occurs in these simulations. In other situations, such as in a nonlinearly unstable plasma when the amplitude is growing, non-perturbative trapping scenarios can come into play as well giving rise to Gaussian like SEHs [13,14].

In the nonlinear Vlasov regime, advances in the construction of hole equilibria went hand-in-hand with the abandonment of the Bernstein, Greene, and Kruskal method (BGK method [31]) as a reliable method, since the necessary phase velocity as an essential element of a theoretical description cannot be given by it. The progress in theory is instead based on the use of Schamel's pseudo-potential method [11], the only method that can provide complete, self-consistent solutions to the VP system.

Therefore, the focus of the new innovations is on the various trapping scenarios that occur during the evolutionary process and are physically caused by events, such as phase space folding, filamentation, ballistic motion, trapping, detrapping or retrapping etc. As said, the trapping scenarios will be mainly perturbative for small amplitudes. When the amplitude is growing also non-perturbative ones, existing for non-zero amplitudes only, can come into play during the evolution and contribute

additively in an asymptotic settled equilibrium solution. These plasma phenomena are hence intimately connected with the chaotic behavior of a single particle in its resonant interaction with a coherent wave and, hence, with its stochastic motion in the phase space region where the discrimination between free and trapped particles takes place. Therefore, collective particle trapping has many faces that need to be explored to obtain a broader view of structure formation in collision-free plasmas.

In this paper, we focus on an intrinsic deterministic math problem, namely how the shape $\phi(x)$ and the phase velocity $v_0$ of a basic solitary electron hole are affected by the simultaneous presence of additional trapping channels.

We treat the most familiar binary trapping systems that are characterized by the two limiting solitary wave solutions, the $sech^4(x)$ solitary wave and the Gaussian $e^{-x^2}$ solitary wave, and evaluate, in the first part, the shape and its corrections. In the second part, we consider the phase velocity and its dependence on a possible third trapping scenario and work out the corrections that are associated with it. The third part deals with an evolution equation of Generalized Schamel type, which is suitable for describing the behavior of the generalized two-parametric SEH in space-time.

## 2. The Basics of the Pseudo-Potential Method

To describe the desired effects as transparently as possible we study the simplest possible plasma, a two-component, current-less plasma with immobile ions and unperturbed Maxwellian electrons, i.e., we focus on electron trapping effects for SEHs propagating in the electron thermal range. Therefore, we start with a stationary solution of the electron Vlasov equation, $(v\partial_x + \phi'(x)\partial_v)f_e(x,v) = 0$, in the wave frame where the structure is at rest, given by (1) in [13], which reads

$$f_e(x,v) = \frac{1}{\sqrt{2\pi}}\left(\theta(\varepsilon)\exp[-\frac{1}{2}(\sigma\sqrt{2\varepsilon} - v_0)^2] + \right.$$
$$\left. \theta(-\varepsilon)\exp(-\frac{v_0^2}{2})\{1 + [\gamma + \chi\ln(-\varepsilon)]\sqrt{-\varepsilon} - \beta\varepsilon\}\right). \tag{1}$$

In this equation, $\theta(x)$ represents the Heavyside step function, $\varepsilon := \frac{v^2}{2} - \phi(x)$ is the single particle energy and $v_0$ is the phase velocity in the electron lab frame. We use normalized quantities such that the velocity is normalized by the (unperturbed) electron thermal velocity, the electron potential energy by the electron thermal energy, and the space by the Debye length.

Its form results from the Galilei transformation shift $v_0$ of the Maxwellian given in the unperturbed case by $f_M(v) = \frac{1}{\sqrt{2\pi}}\exp(-\frac{(v-v_0)^2}{2})$ and from the replacement of $v$ by $\sigma\sqrt{2\varepsilon}$ as an effect of the perturbation, where $\sigma := v/|v|$ is the sign of the velocity. Notice that $f_e(x,v)$ is thus a function of two constants of motion, $\epsilon$ and $sgv$, both being a necessary requisite for a propagating wave solution. It consists of two parts, the contribution of untrapped particles, $\varepsilon > 0$, and the one of trapped particles, $\varepsilon \leq 0$. Hence, trapping is controled by the three parameters $\beta$, $\gamma$ and $\chi$, the first two refer to a perturbative treatment of trapped particle effects and represent the first two elements of a Taylor expansion with respect to $\sqrt{-\varepsilon}$ whereas the third one, $\chi$, is definitely non-perturbative in nature. Note that $f_e(x,v)$ is continuous across the separatrix and it is assumed that $0 \leq \phi(x) \leq \psi << 1$.

Here, we interrupt to say the following: the whole world of collective trapping is represented by the bracket $\{...\}$ of $f_{et}$ in (1) and the manifold of functions being possible there. Here, we restrict the analysis to three terms represented by the parameters $\gamma, \beta, \chi$ and justify this by its plausibility and the reduction to known cases but we admit that other terms such as $\chi_n \ln^n(-\varepsilon)\sqrt{-\varepsilon}$ with fractional powers of $n < 3$, including $n = 2$, could also be added ([14]). This is the interface between discrete particle physics, the trajectories of which coincide with the characteristics of the Vlasov equation, and the collective particle trapping physics in the mean field description, a largely unsolved problem.

To get a physical idea, consider the situation of an increasing amplitude linearly triggered by a broad band packet of waves with random phases and / or nonlinearly by seeds. In course of time, due to the filamentary fragmentation of the distribution, a number of trapping events may

take place one after the other that asymptotically accumulate in the considered perturbative and/or non-perturbative trapping scenarios, respectively.

The electron density $n_e(\phi)$ can either be obtained by the velocity integration of (1) and subsequent velocity integration, while using $\phi << 1$, as done in [11,20–22] or by the Taylor expansion of (1) first, followed by the velocity integration, as done in [19,32,33]. In both cases, the result is:

$$n_e(\phi) = 1 + \left[A - \frac{1}{2}Z_r'(\frac{v_0}{\sqrt{2}})\right]\phi - \frac{5B}{4\sqrt{\psi}}\phi^{3/2} + D\phi\ln\phi \tag{2}$$

where $A := \frac{\sqrt{\pi}}{2}[\gamma + \chi(1 - 2\ln 2)]e^{-\frac{v_0^2}{2}}$, $B := \frac{16}{15}b(\beta, v_0)\sqrt{\psi}$ with $b(\beta, \tilde{v}_D) := \frac{1}{\sqrt{\pi}}(1 - \beta - v_0^2)e^{-v_0^2/2}$ and $D := \frac{\sqrt{\pi}}{2}e^{-\frac{v_0^2}{2}}\chi$. By introducing a new notation for the trapping parameter $\gamma$, defined by $\Gamma := \frac{\sqrt{\pi}}{2}e^{-\frac{v_0^2}{2}}\gamma$, we obtain $A = \Gamma + (1 - \ln 4)D$.

After insertion of the density into Poisson's equation, $\phi''(x) = n_e - 1 =: -\mathcal{V}'(\phi)$, where, in the last step, the pseudo-potential $\mathcal{V}(\phi)$ has been introduced, we get

$$-\mathcal{V}'(\phi) = \left[A - \frac{1}{2}Z_r'(\frac{v_0}{\sqrt{2}})\right]\phi - \frac{5B}{4\sqrt{\psi}}\phi^{3/2} + D\phi\ln\phi \tag{3}$$

and by integration with $\mathcal{V}(0) = 0$

$$-\mathcal{V}(\phi) = \frac{\phi^2}{2}\left(\left[A - \frac{1}{2}Z_r'(\frac{v_0}{\sqrt{2}})\right] - B\sqrt{\frac{\phi}{\psi}} + D(\ln\phi - \frac{1}{2})\right) =: -\mathcal{V}_0(\phi). \tag{4}$$

In (4), we have introduced a subscript 0 in $\mathcal{V}_0(\phi)$ in order to indicate that this is a preliminary function. The necessary constraint of a second zero of $\mathcal{V}_0(\phi)$, at $\phi = \psi$, yields (5), the nonlinear dispersion relation (NDR):

$$\left[A - \frac{1}{2}Z_r'(\frac{v_0}{\sqrt{2}})\right] = B + D[\frac{1}{2} - \ln\psi] \tag{5}$$

which can be understood as an implicit function of $v_0$ in dependence of (A, B, D, and $\psi$). Inserting (5) into (4), we get the final version of $\mathcal{V}$, as given by

$$-2\mathcal{V}(\phi) = B\phi^2(1 - \sqrt{\frac{\phi}{\psi}}) + D\phi^2\ln(\frac{\phi}{\psi}) \tag{6}$$

Note that, in this last step, no knowledge of $\phi(x)$ is needed. The latter is obtained by a quadrature of the pseudo-energy as will be demonstrated later (see (7) and (8)). We emphasize that both (5) and (6) are necessary requisites for a complete nonlinear wave theory, i.e., only through $\phi(x)$ from (6) and $v_0$ from (5) the final wave solution $\phi(x - v_0 t)$, $f_e(x - v_0 t, v)$ can be obtained. If it turns out that within the pseudo-potential method a $v_0$ can not be found, the chosen $f_{et}(-\varepsilon)$-ansatz was taken too narrowly implying that further trapping scenarios, i.e., further trapping parameters, have to be incorporated in order to find a $v_0$.

It should also be noted that, within the BGK method [31], which is equivalent to the first step here, there is no way of getting a $v_0$! The reason is that, in (6), which in generalized form contains the same amount of information that is needed for the BGK method, the $v_0$-dependency has dropped out. Nevertheless, there seems to be a predominant opinion in the literature that the BGK method is more general than the pseudo-potential method. This is definitely not true. The BGK method provides a one-to-one correspondence between $\phi(x)$ and $f_{et}(-\varepsilon)$ i.e., either one of these quantities can be used to

describe completely the shape of the potential. Hence, the manifold of $\phi(x)$ is uniquely mirrored in the manifold of $f_{et}(-\varepsilon)$ and vice versa. Accepting that any $f_{et}(-\varepsilon)$ is admitted in the pseudo-potential method, the generality of the BGK method is hence transferred to the one of the pseudo-potential method. This refers however to the shape only and involves at this stage also BGK solutions that are unphysical or don't possess a phase velocity, an issue that can only be circumvented by applying the pseudo-potential method.

We add that there is also a generic argument in favor the the pseudo-potential method, as $\phi(x)$ is a derived quantity, whereas the distribution is an intrinsic one being determined by internal microscopic processes. This implies that $f_{et}(-\varepsilon)$ is the primary function from which $\phi(x)$ is obtained and not vice versa. Moreover, as we will see, it is by no means obvious that a $\phi(x)$ is established that can be written explicitly in terms of known functions, a necessary restriction for the use of the BGK method.

Moreover, by means of (5), we can obtain a simpler notation for the electron density that is given by

$$n_e(\phi) - 1 = B\,\phi(1 - \frac{5}{4}\sqrt{\frac{\phi}{\psi}}) + D\phi[\ln\frac{\phi}{\psi} + \frac{1}{2}] = -\mathcal{V}'(\phi) \tag{7}$$

## 3. The Potential $\phi(x)$ and Its Lack of Analytical Disclosure

We now turn our attention to the first step, the shape of a SEH that is determined within the pseudo-potential method by the pseudo-energy

$$\frac{\phi'(x)^2}{2} + \mathcal{V}(\phi) = 0, \tag{8}$$

where $\mathcal{V}(\phi)$ is the pseudo-potential in its canonical form, as given by (6), in which $D < 0$ and $B > 0$ characterize the strengths of the considered two trapping scenarios in isolation being related with $\chi$ and $\beta$, respectively ([12,13]).

Hence, both basic SEHs (without the second trapping effect) are given by: $\phi(x) = \psi e^{Dx^2/4}$ and $\phi(x) = \psi\,\text{sech}^4(\frac{\sqrt{B}x}{4})$, respectively.

Utilizing (6) and (8), we get, for the implicit shape of $\phi(x)$, the following expression:

$$x = -\int_\psi^\phi \frac{d\tilde{\phi}}{\sqrt{-2\mathcal{V}(\tilde{\phi})}} = \int_{\phi/\psi}^1 \frac{d\xi}{\xi\sqrt{D\ln\xi + B(1 - \sqrt{\xi})}}, \tag{9}$$

i.e., to get $x = x(\phi)$ (or by inversion $\phi = \phi(x)$) we "simply" have to perform the integration.

However, the problem we are faced with is that for arbitrary $D, B$ a solution of (9) cannot be found in terms of known standard functions (WolframAlpha). This does not imply that a two-parametric SEH is non-existent, but merely that this generalized structure cannot be expressed anymore by reference to familiar functions. Nevertheless, to obtain an idea regarding the effect of a second channel of trapping, we have to address both SEHs separately and assume the second trapping mechanism to be weak.

(i) Modified Gaussian

In case of $\frac{B}{|D|} =: \epsilon \ll 1$ we Taylor expand (9) to get

$$\sqrt{-D}x = \int_{\phi/\psi}^1 \frac{d\xi}{\xi\sqrt{-\ln\xi}} - \frac{\epsilon}{2}\int_{\phi/\psi}^1 \frac{d\xi(1 - \sqrt{\xi})}{\xi(-\ln\xi)^{3/2}}. \tag{10}$$

Both of the integrals in (10) can be solved by means of WolframAlpha to get

$$\sqrt{-D}x = 2\sqrt{-\ln\frac{\phi}{\psi}} + \epsilon\left[1 - \sqrt{\frac{\phi}{\psi}} - \frac{\sqrt{\pi}}{2}\,\text{erf}(\sqrt{\frac{-\ln\frac{\phi}{\psi}}{2}})\right] \tag{11}$$

As can be seen by inspection, the term after $\epsilon$ in (11) is negative for given $\phi$. This means that $x(\phi) > 0$ becomes smaller and, hence, the potential $\phi(x)$ as a function of x narrower, i.e., the width of $\phi(x)$ shrinks as an effect of the secondary trapping. This can also be seen by the inversion of (11) which reads

$$\phi(x) = \psi e^{Dx^2/4} \left[ 1 + \epsilon \frac{\sqrt{-D}x}{4} [1 - e^{Dx^2/8} - \frac{\sqrt{\pi}}{2} \operatorname{erf}(\frac{\sqrt{-D}x}{2\sqrt{2}})] \right] \tag{12}$$

in which the term behind $\epsilon$ is negative. Note that the effect of $B$, i.e., of the second channel, is hidden in $\epsilon$.

(ii) Modified $sech^4$ solitary electron hole

Next, we investigate the effect of a weak, non-perturbative Gaussian-type trapping process on the "regular" $sech^4$ solitary electron hole. In this case, we have to solve

$$\sqrt{B}x = \int_{\phi/\psi}^{1} \frac{d\xi}{\xi\sqrt{(1 - \sqrt{\xi}) + \epsilon(-\ln\xi)}} \approx \int_{\phi/\psi}^{1} \frac{d\xi}{\xi\sqrt{(1 - \sqrt{\xi})}} + \frac{\epsilon}{2} \int_{\phi/\psi}^{1} \frac{d\xi(-\ln\xi)}{\xi(1 - \sqrt{\xi})^{3/2}}, \tag{13}$$

where $0 \le \epsilon := -D/B << 1$ and where the Taylor expansion has already been made. Both of the integrals can be performed. The first one leads to $-4\tanh^{-1}\left(\sqrt{1 - \sqrt{\phi/\psi}}\right)$ from which follows by inversion our basic "regular" SEH: $\phi(x) = \psi\operatorname{sech}^4\left(\frac{\sqrt{B}x}{4}\right)$. However, the second integral assumes the very tedious form (WolframAlpha)

$$\left[ \frac{0.444444}{(1 - \sqrt{\xi})^{3/2}\sqrt{\xi}} \sqrt{1 - 1/\sqrt{\xi}} \left( 4(\sqrt{\xi} - 1)_3F_2(1.5, 1.5, 1.5; 2.5, 2.5; \frac{1}{\sqrt{\xi}}) - 9(\sqrt{\xi} - 1)\xi^{3/4}\ln\xi\sin^{-1}(\xi^{-1/4}) + 9\sqrt{1 - \xi^{-1/2}}\xi\ln\xi \right) \right]_{\phi/\psi}^{1}, \tag{14}$$

such that a further progress is terminated here. In (14), $_3F_2(a_1, a_2, a_3; b_1, b_2; x)$ is a generalized hypergeometric function for which even in the upper case $\xi = 1$ no explicit, manageable expression seems to exist. There exists a Taylor expansion with combinations of Gamma functions $\Gamma(k + \frac{3}{2})$ and $\Gamma(k + \frac{5}{2})$, $k = 0, 1, 2, 3, ...$ as coefficients, but we are completely lost if we take the lower case $\xi = \phi/\psi$.

We have to accept that we have reached in this special situation mathematical treatability. The addition of a weak, non-perturbative Gaussian trapping process on the regular one has the detrimental effect of analytical non-tractability. We have left the region in which physically deterministic processes can be analytically treated using standard functions.

One may object that a Taylor expansion with respect to $\epsilon(-\ln\xi)$ breaks down at $\xi = 0, \phi = 0$ such that (14) is not, seriously speaking, justified. If this is correct, then (14) is more an illustration of how challenging it is to express $\phi(x)$ as a solution of (9). Nevertheless, this does not change our argumentation that our binary trapping system (D,B) has no solution $\phi(x)$ that can be expressed by standard functions. Although $\phi(x)$ **does** exist as a bell-shaped function, but it remains analytically unresolved. One way to get its shape is to numerically perform the integral. Because, on the other hand, a known $\phi(x)$ is needed for the application of the BGK method [31], the limit of applicability of the latter is reached in dealing with this problem. This is a further reason why the pseudo-potential method is preferable to the BGK method.

Therefore, our main conclusion is that the diversity of SEHs, as mentioned at the beginning, not only refers to analytically expressible solutions, but involves structures, as well, with a hidden mathematical background of solutions. The complexity of trapping does not make any difference between mathematically expressible or non-expressible solutions, if we want to or not.

An interesting observation is that $\phi(x)$ can be expressed in the vicinity of the non-perturbative Gaussian, but not in the vicinity of the perturbative sech$^4$ SEH. The Gaussian SEH seems to be more robust to perturbations than the *sech*$^4$ SEH, at least within the considered class.

Moreover, we may point out that this non-expressionality of $\phi(x)$ cannot be automatically transferred to other binary trapping scenarios. As a counterexample we consider the two non-perturbative trapping nonlinearities in the trapped electron distribution: $\chi_1 \ln(-\varepsilon)\sqrt{-\varepsilon}$ and $\chi_2 \ln^2(-\varepsilon)\sqrt{-\varepsilon}$. The potential is given for any $\chi_1$ and $\chi_2$ by $\phi(x) = \psi e^{-sX(x)^2}$, where $X(x) = \sinh(\sqrt{D_2}x/2)$, where $s$ and $D_2$ are related with $\chi_{1,2}$ and $\psi$, as shown in [14]. In this case, a presentable solution exists in the whole range spanned between these two isolated Gaussian-type holes.

As $\mathcal{V}(\phi)$ is the more fundamental of the two $(\mathcal{V}, \phi)$, we can already decide on the $\mathcal{V}$ level as to whether SEHs exist without knowing the explicit x-dependency of $\phi$. Namely, by demanding $\mathcal{V}''(0) < 0$, we obtain the necessary constraint $-\mathcal{V}''(0) = \lim_{\phi \to 0}[B(1 - \frac{15}{8}\sqrt{\frac{\phi}{\psi}} + D(\ln\phi + 1)] = \lim_{\phi \to 0}(D \ln \phi) > 0$ and hence $D < 0$, i.e., a negative $D$ guarantees the existence of a solitary wave even if its explicit x-dependence remains undetermined.

The explicit knowledge of $\phi(x)$ as a prerequisite for the applicability of the BGK method also applies to the derived quantity, the bipolar structure, a central structure that is omnipresent in space observations. It refers to the electric field $E(x) = -\phi'(x)$, which is given by $E(x) = \pm\sqrt{-2\mathcal{V}(\phi)}$. Or, using $x = x(\phi(x))$, we get by differention the equivalent expression $E(x) = -\frac{1}{x'(\phi(x))}$. This means that the lack of analytical disclosure is also transferred to $E(x)$, and it holds as long as the functional dependence of $\phi(x)$ is unknown.

## 4. The Phase Velocity $v_0$

Next, we look at the second important part of a nonlinear wave solution, the phase velocity $v_0$. Because this is not accessible by the BGK method, many aspects of this topic remained unknown to the majority of the plasma community. We want to analyze the NDR (5), which we write as

$$-\frac{1}{2}Z_r'(\frac{v_0}{\sqrt{2}}) = B + D[\frac{1}{2} - \ln\psi] - A = B - \Gamma - D[\frac{1}{2} - 2\ln 2 + \ln\psi] =: \mathcal{R}, \tag{15}$$

where in the second definition of $\mathcal{R}$ the three trapping scenarios $(\Gamma, D, B)$ or $(\gamma, \chi, \beta)$ are well distinguished.

In this equation, the parameters $B$ and $D$, which determine $\mathcal{V}(\phi)$ and are the source of the trouble with the explicit shape of $\phi(x)$, are thought to be given. It is the parameter $A$ (or $\gamma$, respectively) that provides solutions of (15) in a wide, continuous range of $v_0$.

An inspection of $-\frac{1}{2}Z_r'(\frac{v_0}{\sqrt{2}})$, see e.g., Figure 3 of [10], shows that a *slow* solution branch $0 \le v_0 \le 2.12 = 1.5\sqrt{2}$ exists when $1 \ge \mathcal{R} \ge -0.285$ and a *fast* branch, $2.12 \le v_0$, when $-0.285 \le \mathcal{R} \le 0$. The latter branch is reminiscent of the Langmuir branch and it is obtained for $\mathcal{R} \to 0^-$, i.e., in the fluid limit [24]. The second zero of $\mathcal{R}$ refers to $v_0 = 1.307$, a mode that also formally exists as a linear mode, but that, in contrast to the common belief, can only be understood correctly as a nonlinear mode [18,25]. It is the *slow electron acoustic wave* on which the classical SEH solution $(D = 0 = A, B > 0)$ rests [20,24].

An interesting aspect of (15) is the limit $\psi \to 0$. As long as $D \neq 0$ the $\ln\psi$-term in (15) results in an unlimited growth of $\mathcal{R}$ within this limit and, thus, in a violation of the NDR. Non-perturbative SEH solutions of Gaussian type don't have a zero amplitude limit. This regime is reserved for the perturbative, privileged SEHs.

We note in parenthesis that, in the case of a periodic structure, $k_0 \neq 0$, when $k_0^2$ appears additively on the left hand side of (15), $\mathcal{R}$ is often called "nonlinear frequency shift", see [1] and references therein. However, this assignment is problematic, if not misleading. The reason is that there is no linear mode that can be assigned to end in a nonlinear structure that might justify the notion and the introduction of a frequencyy shift. Even if $\mathcal{R} = 0$ the resulting *harmonic* waves are typically nonlinear representing a

specific trapping state (e.g., $B = 0$ or $1 - \beta = v_0^2$, being a *slow mode* ) and not necessarily the Langmuir case $e^{-v_0^2/2} \to 0$ ([18,25]). The functional space of the linear Vlasov modes is detached from that of the non-linear Vlasov modes. There is no cross connection. This especially holds for localized solitary holes. The typical cause for the establishment of a SEH is an initially localized seed fluctuation in $f_e$, which quickly turns into a privileged electron hole ($\beta \neq 0, \gamma \neq 0$), no matter how small the fluctuation. As already said, a privileged SEH does exist for arbitrarily small amplitudes, and, due to $\gamma$ for almost any $v_0$. If it grows in an unstable mode situation, further trapping scenarios take place, which turn it into a more complex SEH with an analytically undisclosed character, the topic of the present paper. Hence, there is no instant where linearity plays any role, such that the term "nonlinear frequency shift" is a questionable abbreviation.

The common view according to Landau and prevailing wave theory is that, in the unstable case with an initial wave packet with random phases and topological perturbations, the most unstable linear mode dominates in course of time getting a given finite wave number. However, during saturation due to trapping, the mode changes its character and undergoes a transit into the non-linear function space, thereby completely forgetting its linear past. Thereby, it not only becomes a nonlinear trapping mode but maintains periodicity, i.e., it cannot transform into a coherent solitary hole with $k_0 = 0$, the most observed structure. Or, generally speaking, coherency, and especially the generation of localized structures, are in conflict with linear Landau theory, which hence fails to be a universally valid theory [8,24–27,33–36].

Another example of the misuse of linear wave concepts in the nonlinear wave regime is the *group velocity* in Rayleigh's sense: $v_g := \partial\omega/\partial k$ [37–41]. Because there is no linear carrier wave that can be assigned, a nonlinear wavelet, being understood as a spatially shortened solution of cnoidal holes, propagates as a bulk with $v_0$, instead, the associated phase velocity [27]. "It is therefore by no means surprising when authors in [42,43] find strong deviations of the group velocity as a result of particle trapping and conclude that the group velocity $v_g$ of an essentially undamped wave, calculated by using the very definition of Rayleigh, is found to significantly differ from $\partial\omega/\partial k$ or that surprisingly enough the main nonlinear change in $v_g$ occurs once the wave is effectively undamped".

We now continue with the further discussion of the NDR (15).

When $|\mathcal{R}| << 1$, we can solve (15) using $-\frac{1}{2}Z_r'(x) \approx -\frac{1}{x_0}(x - x_0)$, $|x - x_0| << 1$, $x_0 = 0.924$, ref. [24], to obtain

$$v_0 = 1.307(1 - \mathcal{R}). \tag{16}$$

Hence, the phase velocity $v_0$ is sub (super) critical with respect to the *slow electron acoustic velocity* $C_{SEA} := 1.307$ if $\mathcal{R} > 0$ ($\mathcal{R} < 0$).

Expressed by the trapping parameters $(\Gamma, D, B)$, $v_0$ becomes: $v_0 = 1.307\left(1 - B + \Gamma + D(\frac{1}{2} + \ln\frac{\psi}{4})\right)$.

An important aspect now is that, for every meaningful doublet $(B, D)$ that specifies $\mathcal{V}(\phi)$, the third trapping mechanism $\Gamma$ can always be adjusted, so that $v_0$ is placed anywhere in $0 < v_0 < \infty$. In other words, although the explicit form of $\phi(x)$, depending on $(B, D)$, can be analytically undetermined, the $\gamma$-trapping process can always be determined, so that a predetermined phase speed is reached. Therefore, there is a high degree of flexibility in the three trapping processes to establish a SEH e.g., near $C_{SEA}$.

## 5. The Nonlinear Evolution Equation

The essence of Schamel's pseudo-potential method is that it alone gives access to the two main criteria for the completeness of a nonlinear wave solution: the existence of :

(i)  a pseudo-potential $\mathcal{V}(\phi)$ (but not necessarily an explicit $\phi(x)$) and
(ii)  a phase velocity $v_0$.

In our present case, $\mathcal{V}(\phi)$ is given by (6) and $v_0$ by (16) with $\mathcal{R}$ given by (15) for structures that propogate near $C_{SEA}$.

Our goal here is to establish an evolution equation that describes this mode in the stationary limit $\phi(x,t) = \phi(x - v_0 t)$, but enables more complex space-time dependent processes, such as the collision of two members.

The simplest way to get such an equation is to use and combine both concepts, assuming that, in the stationary limit, the evolution equation decomposes into $-\mathcal{V}''(\phi)\phi_x - \phi_{xxx} = 0$, which is obtained by an x-differention of Poisson's equation, $\phi_{xx} = -\mathcal{V}'(\phi)$, and into $\phi_t + v_0\phi_x = 0$ with $v_0$ given by (16) and $\mathcal{R}$ given by (15). Accordingly, we come to $\phi_t + v_0\phi_x + 1.307[-\mathcal{V}''(\phi)\phi_x - \phi_{xxx}] = 0$, where we used the coupling constant $C_{SEA} = 1.307$. By inserting $-\mathcal{V}''(\phi)$ using (6), i.e., $-\mathcal{V}''(\phi) = B(1 - \frac{15}{8}\sqrt{\frac{\phi}{\psi}}) + D(\ln\frac{\phi}{\psi} + \frac{3}{2})$, and $v_0$ from (16) with $\mathcal{R}$ given by (15), we hence obtain:

$$\phi_t + 1.307\left[\mathcal{A} - B\frac{15}{8}\sqrt{\frac{\phi}{\psi}} + D\ln\frac{\phi}{\psi}\right]\phi_x - 1.307\phi_{xxx} = 0 \tag{17}$$

with $\mathcal{A}$ given by

$$\mathcal{A} = 1 + A + D(1 + \ln\psi) = 1 + \Gamma + D(2 + \ln\frac{\psi}{4}) \tag{18}$$

and $A, B, D, \Gamma$ being defined after (2). (17) is our desired evolution equation, which, by construction, satisfies (6) and (9) with $v_0$ given by (16).

A still more attractive form is obtained if we use the rescaled variables $\varphi := \phi/\psi$ and $\tau := t/C_{SEA}$ to obtain

$$\varphi_\tau + [\mathcal{A} - B\frac{15}{8}\sqrt{\varphi} + D\ln\varphi]\varphi_x - \varphi_{xxx} = 0. \tag{19}$$

For more details to further justify the derivation and robustness of the equation, see Appendix A. Next, we divide the discussion into the perturbative and partially non-perturbative trapping limit.

In the perturbative trapping limit, we get by setting $D = 0 = \chi$ from (17)

$$\phi_t + 1.307\left[1 + A - B\frac{15}{8}\sqrt{\frac{\phi}{\psi}}\right]\phi_x - 1.307\phi_{xxx} = 0 \tag{20}$$

with $A = \Gamma = \frac{\sqrt{\pi}}{2}\gamma e^{-\frac{C_{SEA}^2}{2}} = 0.378\gamma$. With $\frac{15}{8}B = 2b\sqrt{\psi}$ we hence get

$$\phi_t + 1.307\left[1 + \Gamma - 2b\sqrt{\phi}\right]\phi_x - 1.307\phi_{xxx} = 0 \tag{21}$$

This is identical to the Schamel equation (Equation (15) from [22]), which appears here in an extended form due to the perturbative trapping term $\Gamma$, which was not considered at this early stage in [22]. With the participation of $\Gamma$, the phase speed, as given (16), becomes more generally $v_0 = C_{SEA}(1 - B + \Gamma)$.

It should be noted that the original Schamel equation (formerly referred to as the modified Korteweg de Vries equation) was first derived in [32], namely Equation (15), for solitary ion sound waves with non-isothermal electrons. This regime is equivalent to SEHs derived for structures propagating more slowly in the ion acoustic regime, $v_0 = \sqrt{m_e/m_i} =: C_S$, which however requires mobile ions, i.e., a finite ion mass $m_i$, the coupling constant being given by $C_S$ in this case.

With the second, analytically feasible option, the non-perturbative trapping limit, $B = 0$, but keeping the perturbative $\Gamma$ trapping term, we are breaking new ground. We obtain

$$\phi_t + 1.307\left[1 + \Gamma + 2D(1 - \ln 4) + D \ln \phi\right]\phi_x - 1.307\phi_{xxx} = 0. \tag{22}$$

This represents the Gaussian limit of (17). This limit is outside the scope of a BGK analysis, because it contains the concept of a phase velocity. Hence, its solitary wave solution is given by $\phi(x,t) = \psi e^{D(x-v_0 t)^2/4}$ with $v_0 = 1.307\left(1 + \Gamma + D(\frac{1}{2} + \ln\frac{\psi}{4})\right)$. As before, due to the $\Gamma$ trapping mechanism, there is a high dynamic flexibility of the *Gaussian* SEH in order to achieve a speed around $C_{SEA}$.

Equation (17) in its general form represents a new evolution equation that we may term the "Generalized Schamel"—equation and abbreviate it as the GS equation. The new feature of (17) is that, for non-zero values of $B, D$, a stationary solution $\phi(x - v_0 t)$ can no longer be analytically expressed, albeit a $v_0$ could be assigned. This is due to the missing step to describe $x(\phi)$ and especially its inversion $\phi(x)$ by standard mathematical functions, as discussed in Sect.III. There is access to the speed $v_0$, but not to the explicit form $\phi(x)$ of the structure.

We conclude that the $(B, D)$ trapping scenarios generally imply an area in the function space of solutions where there is no longer any analytical accessibility, a previously unknown property of non-linear wave solutions.

In Appendix B, we provide proof of a further extension of the GS equation by taking a second order Gaussian trapping scenario into account.

## 6. Summary and Conclusions

The paper, which to some extent completes the interim conclusion of a long-term development and reveals further details of the mystery of coherent phase space vortices, has two main achievements:

(i)  the simultaneous presence of the two main trapping scenarios, the perturbative $\beta$- and non-perturbative $\chi$- scenario, leads to a two-parametric solitary wave electric potential $\phi(x)$ (represented by $\mathcal{V}(\phi)$), which, however, can no longer necessarily be analytically described by the help of standard mathematical functions anymore, and

(ii)  a new evolution equation for $\phi(x,t)$ was nevertheless proven which describes its nonlinear, weakly time-dependent dynamics in the vicinity of equilibria.

Two major factors contributed to this innovation, the use of the pseudo-potential method and the existence of both a pseudo-potential $\mathcal{V}(\phi)$ and a phase velocity $v_0$, the latter satisfying a nonlinear dispersion relation (NDR). As a result, the Schamel term $\sqrt{\phi}\phi_x$ in the ordinary Schamel equation, which stems from the $\beta$-trapping scenario, gets a companion $\ln \phi \phi_x$, which arises from the Gaussian trapping process. It will certainly be of great interest to learn how the dynamics is controlled by both terms, a process that can probably only be numerically solved. To simplify access to the world of collective particle trapping, we limited ourselves to the trapping of electrons by considering immobile ions ($\delta := \frac{m_e}{m_i} = 0$) and to a current-less plasma assuming a vanishing drift velocity $v_D = 0$ between electrons and ions. Releasing this restriction by using mobile ions, ion trapping, and a two-stream plasma situation leads to an immense increase in diversity of solutions, even in the solitary wave limit ($k_0 = 0$). Hence, by considering the standard trapping parameters, this multitude of solutions is characterized by an at least 13-dimensional parameter space, given by $[\gamma_s, \beta_s, \chi_{1s}, \chi_{2s}; \psi, \theta := \frac{T_e}{T_i}, v_D, \delta, k_0]$, $s = e, i$. This is a minimum, since further trapping scenarios could easily be added and would increase dimensionality, such as $\chi_{\frac{1}{2}s}\sqrt{-\ln(-\varepsilon)}\sqrt{-\varepsilon}$ or $\chi_{\frac{3}{2}s}\sqrt{-\ln(-\varepsilon)}^3\sqrt{-\varepsilon}$ in $\{...\}$ of (1). It would be too optimistic to believe that structure, space-time behavior, and parameter dependency of coherent phase space vortices can be determined while using experimental or numerical techniques.

However, the collision-free Vlasov approach has to be abandoned, even for extremely diluted and hot plasmas in order to capture physical reality, as already mentioned [12–14]. The reason is that one not only has to circumvent the linear singularities of $\delta$-function and principle value type by going into the nonlinear Vlasov regime, but one has also to bypass the cusp-type singularities, which are inevitably present in any stationary, completely nonlinear VP solutions (see (1)).

As pointed out by Korn, Luque, and Schamel [34,44], dissipative hole equilibria in current-driven plasmas unequivocally require the participation of ions in the collective dynamics, as shown within the "primitive" collisional Fokker–Planck description.

In other words, the coherent patches and eddies in intermittently turbulent plasmas cannot be treated correctly without also taking correlations in the resonant particle region into account, meaning that the pure Vlasov–Poisson approximation is too restricted in dealing with such processes adequately.

We conclude with the remark that chaos at the one-particle level has thus left its footprint in collective structure formation. The stochasticity of resonant particle trajectories obviously prevents us from treating particle trapping as a unique, deterministic process in structure formation. We close with the perhaps overly optimistic hope that mathematical explicitity and disclosure may come back through the back door, in the coarse-grained distribution due to the diffusive phase space dynamics that are triggered by self-consistent correlations.

**Funding:** This research received no external funding.

**Conflicts of Interest:** The authors declare no conflicts of interest .

## Appendix A. The Mathematical Robustness of the New Evolution Equation

The modified KdV equation, (15) of [30] (which was later called Schamel equation), was first derived for ion acoustic, solitary - like wave structures using the reductive perturbation method and taking into account electron trapping effects that deviate from isothermality. A simpler construction with the same result (as long as $D = 0$) is to look for a nonlinear evolution equation of the type KdV ( i.e., for which the dispersion is controlled by a $\phi_{xxx}$-term) and which has the solitary electron hole (SEH) of $\text{sech}^4$—type as an equilibrium solution. This can be done either by formulating an equation, $\phi_t + (a_1 + a_2\sqrt{\phi})\phi_x + a_2\phi_{xxx} = 0$, and by adjusting the parameters $a_{1,2,3}$ or by combining the traveling part, $\phi_t + v_0\phi_x = 0$, with the structural shape determining part, the Poisson's equation, or its first derivative respectively, using the underlying acoustic velocity as the appropriate "coupling constant". For SEHs in the ion sound range the latter is given by $C_s$, the ion sound speed, for SEHs in the electron thermal range it is given by the slow electon acoustic speed $C_{SEA}$ (see e.g., Equation (36) of [19]). In the present two parametric case, when the shape $\phi(x)$ (together with its width) is mathematically undisclosed (and lacking in order to justify a reductive perturbation method) only the second procedure is left. The derivative of Poisson's equation is hence suggested by the dispersive part in a KdV -type equation. Moreover, our new Equation (17) has the correct $D = 0$ limit.

In a mathematical context, one should note that strong convergence holds macroscopically i.e., that the evolution equation for the macroscopic $\phi(x, t)$ should be rather insensitive to fluctuations and hence robust to microscopic changes.

## Appendix B. Extension of the GS Equation by a Second Order GAUSSIAN Trapping Scenario

In this Appendix B we present a further extension of the GS equation by allowing a second order non-perturbative Gaussian trapping scenario. This is done by replacing $\chi$ in (1) (or $D$ in (17)) by the two-Gaussian trapping processes $(\chi_1, \chi_2)$ (or $(D_1, D_2)$, respectively) in line with the investigations of [14].

This extension is introduced by a replacement of $\{...\}$ in (1) by

$$\{1 + [\gamma + \chi_1 \ln(-\varepsilon) + \chi_2 \ln^2(-\varepsilon)]\sqrt{-\varepsilon} - \beta\varepsilon\}. \tag{A1}$$

In other words, our previous Gaussian-like trapping scenario, now labeled Index 1, receives a second-order Gaussian competitor, so that altogether four trapping scenarios are involved.. The application of the previous analysis to this new situation, which we leave to the reader for confirmation, yields

$$-2\mathcal{V}(\phi) = \phi^2 \left[ B(1 - \sqrt{\frac{\phi}{\psi}}) + (D_1 + \hat{r}D_2)\ln(\frac{\phi}{\psi}) + D_2\ln^2(\frac{\phi}{\psi}) \right].\tag{A2}$$

This is our new pseudo-potential, where $D_{1,2} := \frac{\sqrt{\pi}}{2}e^{-v_0^2/2}\chi_{1,2}$ and $\hat{r} := 1 + 2\ln\frac{\psi}{4}$. Using the neighborhood at $C_{SEA}$ of $v_0$ again: $v_0 := 1 - \hat{\mathcal{R}}$, we get

$$\hat{\mathcal{R}} := B - \Gamma - D_1[\frac{1}{2} + \ln\frac{\psi}{4}] - D_2[-\frac{5}{2} + \ln 4(\ln 4 - 1) + \frac{\pi^2}{3} + (1 - 2\ln 4)\ln\psi + \ln^2\psi],\tag{A3}$$

an expression which coincides in the limit of $D_2 = 0$ with $\mathcal{R}$ of (15). Having the two necessary formulas (for $\mathcal{V}$ and $v_o$) we can repeat the previous procedure to get:

$$\phi_t + 1.307\left[\hat{\mathcal{A}} - B\frac{15}{8}\sqrt{\frac{\phi}{\psi}} + (D_1 + \hat{r}D_2)\ln\frac{\phi}{\psi} + D_2\ln^2\frac{\phi}{\psi}\right]\phi_x - 1.307\phi_{xxx} = 0,\tag{A4}$$

where $\hat{\mathcal{A}}$ is an extension of $\mathcal{A}$ in (18) and refers to $\Gamma, D_1, D_2$ that may ultimately be performed by the reader.

This is the most general form of the GS equation involving four trapping scenarios $(\gamma, \chi_1, \chi_2, \beta)$.

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
