# Peer review of "Two-Parametric, Mathematically Undisclosed Solitary Electron Holes and Their Evolution Equation"

_plasma, doi:10.3390/plasma3040012_

Round 1

Reviewer 1 Report

Manuscript # plasma-913104

Two-parametric, mathematically undisclosed solitary electron holes and their evolution

By H. Schamel.

In this manuscript the author presents a mathematical model to describe the nonlinear solutions of the Vlasov-Poisson system which are characterized by the emergence of trapping structures in the phase space . This paper is an extension of a recent work (PoP 27, 062302 (2020) ) that now includes several types of trapping of particles

This is a well-researched paper. It is meticulously worked on and presented. In its actual form, I am not sure what the impact of this paper will be, it includes an analytical solution, which is still difficult for non-specialists to understand, and in my view, this methodology seems fine and is well worth publicizing. It appears rigorous and mathematically sound.

I believe the paper contains significant results and should be published after the following matters are dealt with properly. For publication the presentation needs improvement in several respects. Items to be addressed by the author:

1)Given that manuscript is submitted for publication in ”Plasma”, the introduction seems to me a little shor concerning the physical aspects of the problem and the physical context in general: few references to previous works in the field are cited, whether from the analytical ( J.Holloway, J. Dorning, Phys. Rev. A, 44, 3856 (1991); I. Dodin and N. Fisch, Phys. Plasmas 21, 034501 (2014) ) or numerical point of view (T.W. Johnston et al, Phys. Plasmas 16, 042015 (2009); Valentini et al, Phys. Plasmas13, 052303 (2006)). It would also be interesting to explain in more detail the context of electronic acoustic waves, just evoked in the manuscript by the term C_SEA.

2) the statement “ multiple particle trapping events that a plasma can select when forming a structure” is inaccurate. In a Vlasov-Poisson system the only known trapping mechanism is relative to electrostatic trapping, what are the physical trapping mechanisms that are evoked here?

3) The path that leads to the formation of a nonlinear solution (with a hole in the phase space) can be complex and presents different features: vortex fusion, secondary-type  instability of negative masses (Dodin et al) or be facilitated by the existence of a linear wave, such as a linear electronic acoustic wave, which can serve as a seed to facilitate the emergence of a auto-resonance mechanism. Collision-free dissipation can also play a role to obtain the asymptotic solution..

It may then be justified to wonder whether the beta, gamma and ksi parameters of the proposed solution in Eq. (1) do not rather reflect a different accessibility of the same solution or trapping mechanism. The third trapping mechanism proposed by the author remains a bit obscure. Do we really need the three parameters beta, gamma and psi to converge to a "numerical" solution? In other words, what is the dominant mechanism for a given problem?

4) There is a kind of controversy in the literature about how the emergence of the nonlinear solution occurs (giving rise to the emergence of a trapping structure): on the one hand it can emerge from a linear solution (EAW type) or on the other hand this nonlinear solution has no linear counterpart in terms of limit when the electrostatic potential tends towards zero and requires a minimum threshold in the potential to appear.

The solution in beta and gamma seems to be obtained from a perturbative development and seems to correspond to the first category. The second in psi, constructed from a non-perturbative solution, seems to belong to the second category (no equivalence of a linear solution for a low potential). It is not clear a priori whether the two solutions are compatible within the limit where the potential tends towards zero. Is there a potential threshold in the definition of the ksi parameter. ?

5)It is unclear to me the correctness of the statement (page 5) that “ the addition of a weak, non-perturbative Gaussian trapping process on the regular one has the detrimental effect of analytical non –tractability; we have left the region in which physically deterministic processes can be treated analytically using standard functions”.

It has now well-known that the mechanisms ranging from  the linear  collisionless Landau damping till the emergence of nonlinear coherent structure (phase space hole) may be recovered in the same Vlasov-Poisson modelling just be increasing the initial perturbation density term (see the work of Bertrand et al, Nonlinear Phenomena in Vlasov plasmas, Doveil Editions (1989). Thus one might expect the opposite solution: while the distribution function is characterized by the appearance of an increasingly fine filamentation process in the phase space (in the sense of the weak convergence), the potential solution exhibits strong convergence (Villani), it should tend towards a more "accessible" of smoothed solution. One may wonder, at this stage, the importance of collisionless dissipation mechanisms in the accessibility of the solution. The author should comment on this point.

6) I am not entirely convinced about the interpretation given by the author between nonlinear and linear modes in section IV: “During the saturation due to trapping, the mode changes its character and undergoes transit into the nonlinear function space, forgetting completely thereby its linear past.”

it may not be so obvious. First of all, the linear Landau general solution seems to be one of the important points in Villani's work on establishing a general solution. The author's working hypothesis remains valid if one neglects the effect of filamentation or more precisely if one introduces the concept of non-collisional dissipation. However, the reversibility of the Vlasov equation is linked to the process of filamentation of f in the velocity space, which allows us to find the echo mechanism. Thus if the trapping appears in a frequency range where a linear seed exists (Langmuir wave for example or any other linear wave-acoustic geodetic mode in tokamaks or EAW in laser-plasma interaction), the process of frequency drift is well observed in the simulations and verifies well the predictions of Morales and O'Neil for example (for laser-matter interaction). Thus one cannot say that the system has completely forgotten its "linear past".

7) One of the questions remains the observation of the theoretical solution proposed by the author in numerical simulation. It would be interesting to show a Vlasov- Poisson simulation result for the solution proposed by the author, and in particular, if the theoretical solution is numerically stable, if the expected hole velocity is observed numerically or if a hole acceleration process is observed.

8) The author presents several trapping scenarios related to the three parameters beta, gamma and psi. What makes the reading of the paper a bit difficult for non-specialists is the fact it is difficult to associate to each solution or parameter a physical mechanism. Also the author should discuss the physical physical behind each trapping scenario, for the readers to obtain a clearer understanding of the starting model, particularly concerning applications.

9) there are some typographical errors

Topological  page 1, line4

Galileian  page 2,   line 17

Frequency  , page  6, line  10

Reviewer 2 Report

The manuscript "Two-parametric, mathematically undisclosed solitary electron holes and their evolution equation" by H. Schamel, presents an analysis of solitary electron holes produced by three trapping scenarios. It is shown that there are several solutions although they cannot be obtained analytically, but the phase velocity can be obtained in closed form. These results are of significant interest since they extend those previously discussed by the author, based on the pseudo-potential method. At the end of the paper he also gives an evolution equation for the potential based on qualitative arguments although it is not analyzed further.

The paper transmits a clear message that is worth publishing and it is written in an appropriate manner. There are just a few minor clarifications that I feel should be incorporated before publication:

  1. In equation (1) it should be stated what the meaning of sgv is. It seems it refers to the sign of the velocity but it must be written that way.
  2. In pag. 5 after equation (15) there is a reference to this equation but with the wrong number "...that provides solutions of (14)..."
  3. In the next paragraph it is confusing the ranges given for the fast and slow branches. It seems that the interval R>-0.285 pertains to both branches. Please check that.
  4. In Section V, the derivation of the evolution equation is not convincing since it combines the two equations, Poisson's eq. and traveling wave eq. in an unjustified way: they are simply added with a "coupling parameter". Why is the derivative of Poisson's equation taken instead of just the equation itself. It seems the only justification is that a third derivative should be there, as in KdV equation. But this qualitative derivation is not justified. It would be necessary to provide a more convincing argument for the procedure presented.
  5. In pag. 7 there are three citations to references not listed; they are S86 and S73, which I guess they refer to references [11] and [12]. This seems to be a typo in the source text file.
  6. In the Conclusions there is the statement "An incorrigible optimist who believes that structure, space-time behavior and parameter dependency of coherent phase space vortices can be determined using experimental or numerical techniques." It is not understood what is meant to say. Please rephrase.
